# Construction of a Necroptosis-Related lncRNA Signature for Predicting Prognosis and Immune Response in Kidney Renal Clear Cell Carcinoma

**DOI:** 10.3390/cells12010066

**Published:** 2022-12-23

**Authors:** Yue Zhang, Tongtian Zhuang, Zhenlong Xin, Changjian Sun, Deyang Li, Nan Ma, Xiaoyan Wang, Xuning Wang

**Affiliations:** 1Department of Clinical Laboratory, The Air Force Hospital of Northern Theater PLA, Shenyang 110042, China; 2Department of Dermatology, The Air Force Hospital of Northern Theater PLA, Shenyang 110042, China; 3Department of Disease Control and Prevention, The Air Force Hospital of Northern Theater PLA, Shenyang 110042, China; 4Department of General Surgery, The Air Force Hospital of Northern Theater PLA, Shenyang 110042, China

**Keywords:** KIRC, necroptosis, lncRNA, prognosis, immune response, signature

## Abstract

Necroptosis is a new type of programmed cell death and involves the occurrence and development of various cancers. Moreover, the aberrantly expressed lncRNA can also affect tumorigenesis, migration, and invasion. However, there are few types of research on the necroptosis-related lncRNA (NRL), especially in kidney renal clear cell carcinoma (KIRC). In this study, we analyzed the sequencing data obtained from the TGCA-KIRC dataset, then applied the LASSO and COX analysis to identify 6 NRLs (AC124854.1, AL117336.1, DLGAP1-AS2, EPB41L4A-DT, HOXA-AS2, and LINC02100) to construct a risk model. Patients suffering from KIRC were divided into high- and low-risk groups according to the risk score, and the patients in the low-risk group had a longer OS. This signature can be used as an indicator to predict the prognosis of KIRC independent of other clinicopathological features. In addition, the gene set enrichment analysis showed that some tumor and immune-associated pathways were more enriched in a high-risk group. We also found significant differences between the high and low-risk groups in the infiltrating immune cells, immune functions, and expression of immune checkpoint molecules. Finally, we use the “pRRophetic” package to complete the drug sensitivity prediction, and the risk score could reflect patients’ response to 8 small molecule compounds. In general, NRLs divided KIRC into two subtypes with different risk scores. Furthermore, this signature based on the 6 NRLs could provide a promising method to predict the prognosis and immune response of KIRC patients. To some extent, our findings helped give a reference for further research between NRLs and KIRC and find more effective therapeutic drugs for KIRC.

## 1. Introduction

Kidney cancer affects over 430,000 new cases and kills nearly 200,000 individuals worldwide yearly [1], and is the sixth malignancy in men and the ninth in women [2]. Renal cell carcinoma is one of the fatal genitourinary diseases and the most common tumor in the kidney, with kidney renal clear cell carcinoma (KIRC) accounting for about 75% [3]. KIRC patients in the early stages usually have a good prognosis after appropriate treatment. Still, approximately 25% would progress to the advanced stage [4], and a third would suffer a recurrence after the resection of the localized tumor, and the 5-year survival rate is less than 10% [5,6]. Although chemotherapy, immunotherapy, and target therapy have specific therapeutic effects nowadays, the outcome of patients in the late stage remains dismal [4]. Therefore, it is urgent to explore a novel and reliable method to improve the diagnostic accuracy and prognosis for advanced KIRC.

Necroptosis is a new type of programmed cell death mode. American researchers found injecting tumor cells with necroptosis into mice could trigger an anti-tumor immune response. Those cells could guide killer T cells to attack malignant tumors and slow their growth, which means necroptosis may be a promising anticancer strategy [7]. Necroptosis is regulated by multiple genes, especially Mixed Lineage Kinase Domain-Like (MLKL), Receptor-Interacting Protein Kinase 1 (RIPK1), and Receptor-Interacting Protein Kinase 3 (RIPK3) [8]. Many studies have confirmed that necroptosis plays a vital role in different tumors. Up-regulated RIPK3 can remain active in the RIPK3/MLKL signaling pathway and induce necroptosis to alleviate the progression of prostate cancer [9]. Downregulation of RIPK3 promotes lymphatic metastasis in colorectal cancer and results in a poor pathological stage and prognosis [10]. In addition, TRIM28 is an inhibitor in the process of necroptosis. Activated RIPK3 could phosphorylate TRIM28 and increase the production of immunostimulatory cytokine in the tumor microenvironment, then emerge strong cytotoxic anti-tumor immunity [11]. However, the role of necroptosis in tumors is complicated. Necroptosis is an enemy of cancer and its friend [12]. Several studies have found that necroptosis could promote tumor generation, metastasis, and inflammation by mediating the tumor microenvironment [13,14,15]. RIPK3 could suppress CXCL1-induced immunity for promoting colitis-associated colorectal cancer [16], and the release of CXCL5 could aggravate the migration and invasion of pancreatic cancer [17]. Regardless of the dual effects of tumorigenesis and metastasis, necroptosis may be a promising target for tumor therapy.

The long non-coding RNA (lncRNA) usually has more than 200 nucleotides but lacks a protein-coding ability. They can regulate the expression of genes through the lncRNA-miRNA-mRNA axis, thus affecting the phenotype [18]. Growing evidence shows that lncRNA participates in the proliferation, migration, and invasion of various cancers, such as bladder cancer, breast cancer, and lung cancer [19,20,21]. For example, LINC00973 can sponge miR-7109 to control the abundance of Siglec-15 for immune suppression in renal cell carcinoma surface [22]. There are few studies on the necroptosis-related lncRNA (NRL) in KIRC, and further investigation to predict the prognosis of KIRC is warranted.

In this research, we constructed a novel signature based on 6 NRLs by mining the transcriptome and clinical data from TGCA-KIRC and validated their predictive value in different risk groups. We hope this study can provide a more effective diagnostic and therapeutic strategy and improve prognostic outcomes for KIRC patients.

## 2. Materials and Methods

### 2.1. Data Preparation and Cleaning

We downloaded the transcriptome and clinical data from the TCGA-KIRC dataset, then deleted the data with missing information and a short survival time (<30 d). Finally, we got a cohort including the sequencing and survival data of 509 KIRC patients. The cohort was randomly divided into two parts via the “caret” package for subsequent internal validation (Table 1). Besides this, we collected 264 genes involved in the necroptosis pathway from three websites (GeneCards, KEGG, and NCBI), 258 of which were in this transcriptome data (Appendix A). This study defined these 258 genes as necroptosis-related genes (NRGs).

### 2.2. Identification of NRLs

We selected differentially expressed genes (DEGs) from the 258 NRGs based on |log2FC| > 1 and FDR < 0.05, then performed KEGG and GO analysis through the “clusterProfiler” package. With the annotation file downloaded from the GENCODE website, a total of 13,975 lncRNAs were extracted from the FPKM-standardized transcriptome data. Finally, we calculated the correlation between DEGs and 13,975 lncRNAs via the “limma” package and identified NRLs according to |correlation coefficient| > 0.4 and *p* < 0.001.

### 2.3. Establishment of the Prognostic Signature Based on NRLs

First, the univariate Cox analysis discerned which NRLs were associated with the prognosis. Next, we selected the prognosis-related lncRNA through a minor absolute shrinkage and selection operator analysis (LASSO). Moreover, these candidates from LASSO were used to construct a risk model by multivariate Cox regression analysis. According to the median risk score, each patient in the TCGA-KIRC dataset was calculated the risk score based on this model and classified as high and low risk. Finally, the calculation formula for the risk score is as follows. Coef means the coefficient of each NRL, and X means their expression levels.
(1)$$\mathrm{Risk}\;\mathrm{score}\;=\overset{n}{\underset{i=1}{\sum }}Coef(i)*X\left(i\right)$$

### 2.4. Assessment and Validation of the Prognostic Signature

Kaplan–Meier analysis can compare the survival rate between both risk groups by log-rank test. The Receiver operating characteristic curve (ROC) and the area under the ROC curve (AUC) can evaluate the discrimination of the clinical prediction model:We compared the survival differences between high and low-risk groups.Univariate analysis was applied to determine whether the risk score and other pathological factors, including age, gender, and pathological grade, were related to the prognosis.The multivariate analysis could verify whether risk score is an independent parameter for prognosis.We drew a nomogram to predict the survival rate at 1, 3, and 5 years.

### 2.5. Functional Enrichment Analysis and Immune Response of the Prognostic Signature

The GSEA was applied to reveal the mainly enriched signaling pathways under different risk groups. |NES| > 1 and *p* < 0.05 were the screening threshold for significant pathways. According to GSEA results, we analyzed the infiltration of 16 immune cells and the activities of 13 immune-related pathways for each sample. Besides this, we compared the expression level of usual immune checkpoints in both risk groups. To assess the application of this risk model in response to immunotherapy and chemotherapy, we analyzed the relationship between the risk score and the half-maximal inhibitory concentration (IC50) of chemotherapy drugs used for common cancers by the “pRRophetic” package [23].

### 2.6. Statistical Analysis

All statistical analysis in this study was performed through the RGui and a series of suitable packages. Furthermore, *p* < 0.05 was considered statistically significant.

## 3. Results

### 3.1. Enrichment Analysis of Necroptosis-Related Genes

The detailed flow chart of this research is exhibited in Figure 1. First, we obtained 82 DEGs from 258 necroptosis-related genes, containing 64 up- and 18 down-regulated genes (Figure 2A and Appendix A). Then, we performed the KEGG and GO analyses. The KEGG analysis revealed that necroptosis-related DEGs were mainly in necroptosis, influenza A, neutrophil extracellular trap formation, systemic lupus erythematosus, NOD-like receptor signaling pathway, hepatitis C, lipid and atherosclerosis, measles, Kaposi sarcoma-associated herpesvirus infection, and hepatitis B (Figure 2B). Figure 2C shows the GO analysis results, including biological process (BP), cellular components (CC), and molecular function (MF). In BP, DEGs were mainly enriched in necrotic cell death, programmed necrotic cell death, and cytokine-mediated signaling pathways. In CC, DEGs were in the nucleosome, DNA packaging complex, and protein–DNA complex. In MF, DEGs were in death receptor activity, calcium-dependent phospholipase A2 activity, and protein heterodimerization activity.

### 3.2. Construction of the Prognostic Signature Based on 6 NRLs

We obtained 709 necroptosis-related lncRNAs from the TCGA-KIRC dataset (Appendix A), 181 of which were the prognosis-related lncRNA by univariate Cox regression analysis (Appendix A, *p* < 0.001). Next, LASSO analysis further selected 24 lncRNAs as the candidate. Finally, multivariate Cox regression analysis identified 6 NRLs (AC124854.1, AL117336.1, DLGAP1-AS2, EPB41L4A-DT, HOXA-AS2, LINC02100) from the 24 lncRNAs to establish a signature. Moreover, we showed the expYou didn’t mark hereression level of 6 NRLs in each sample (Figure 3A). Meanwhile, we applied the Cytoscape software to explore the co-expression relationship between the 6 NRLs and necroptosis-related DEGs (Figure 3B, |correlation coefficient| > 0.4). DLGAP1-AS2 was co-expressed with 6 necroptosis-related genes (PLK1, TNFRSF25, STAT2, SLC25A37, IRF9, and EZH2), LINC02100 with BIRC3, STAT4, STAT2, and EZH2, HOXA-AS2 with TNFRSF25, SLC25A37, and EZH2, AL117336.1 with STAT4 and IRF9, AC124854.1 with TLR3, and EPB41L4A-DT with JAK3. As shown in Figure 3C, AC124854.1 and EPB41L4A-DT were protective factors, while AL117336.1, DLGAP1-AS2, HOXA-AS2, and LINC02100 were risk factors. Furthermore, we used the following formula to calculate the risk score for each sample: the risk score = (−0.253 × AC124854.1 expression) + (0.469 × AL117336.1 expression) + (0.681 × DLGAP1-AS2 expression) + (−0.421 × EPB41L4A-DT expression) + (0.388 × HOXA-AS2 expression) + (0.429 × LINC02100 expression).

### 3.3. The Value of the Prognostic Signature

Each patient in the TGCA-KIRC dataset was calculated for their risk score and divided into high-risk and low-risk groups based on the median risk score. Compared with the low-risk group, the 5-year survival rate in the high-risk group was significantly shorter (Figure 4A). All patients’ risk scores in the TGCA-KIRC dataset are shown in Figure 4B. Moreover, as the risk score increased, we found that more and more patients died (Figure 4C). Through univariate Cox regression analysis, we could see clinicopathological factors (age, grade, stage, TNM stage), and the risk score was related to the prognosis of KIRC (Figure 4D).

Moreover, multivariate Cox regression analysis revealed that the risk score was a significant factor that could predict the prognosis independent of other factors. (Figure 4E). In addition, ROC curves showed that the risk score had good power in predictive prognosis and was better than all other clinicopathological features (Figure 4F,G).

Next, we drew a nomogram to further calculate the survival rate at 1, 3, and 5 years for KIRC, including the risk score and other pathological factors (Figure 5A), and the calibration curve showed the predictive survival had a good consistency with the actual OS (Figure 5B–D).

Finally, we also studied the predictive ability of the risk score in many clinicopathological features, including age, gender, and pathological grades. Compared with the low-risk, the OS of the high-risk patient in age, gender, grade, stage, T, and M stage was significantly shorter (Figure 6A–L). All the above results showed that our model could effectively distinguish patients with risk scores.

### 3.4. Internal Validation of the Prognostic Signature

To validate the applicability of our prognostic signature, we randomly divided the entire cohort (*n* = 509) into two validation cohorts (*n* = 256, *n* = 253). Consistent with the results in the entire cohort, the OS of high-risk patients was lower in both cohorts (Figure 7A, *p* = 4.579 × 10^−10^; Figure 7B, *p* = 1.716 × 10^−10^). Besides this, the AUC at 1, 3, and 5 years are, respectively, 0.789, 0.763, and 0.832 in the first internal cohort. The AUC at 1, 3, and 5 years are, respectively, 0.703, 0.749, and 0.795 in the second internal cohort (Figure 7C,D).

### 3.5. Immune-Related Pathways and Immune Microenvironment

We performed the GSEA to explore the difference in gene enrichment between both risk groups. In the high-risk group, the p53 signaling pathway, cytokine–cytokine receptor interaction, and cytosolic DNA-sensing pathway were significantly enriched (Figure 8A–C), indicating that the high-risk group was closely associated with tumor and immune-associated pathways. Then, we quantified the infiltrating level of 16 immune cells and 13 immune functions to explore the difference in the immune microenvironment further. We could find that there were significant differences in CD8+ T cells, DCs, immature dendritic cells (iDCs), macrophages, mast cells, T helper cells, T follicular helper (Tfh) cells, T helper type 1 (Th1) cells, and tumor-infiltrating lymphocytes (TIL) (Figure 9A). Furthermore, the checkpoint, inflammation-promoting, T-cell co-inhibition, and co-stimulation were higher in the high-risk group than in the low-risk group (Figure 9B). These results suggest that in the high-risk group, these immune functions were more active, and the patient might be more sensitive to immunotherapy.

### 3.6. Potential Effects of Immunotherapy and Chemotherapy in KIRC

Since checkpoint blockade is a new immunotherapeutic strategy in cancer, we discussed the relationship between risk score and immune checkpoints. In Figure 10, several classical immune checkpoints had higher expression levels in the high-risk group, such as PDCD-1, CTLA-4, LAG-3, and TIGIT. Furthermore, the TNFRSFs were also higher in the high-risk group, including TNFRSF-8, -9, -14, -18, and -25. Most notably, TNFRSF-25 had a co-expression relationship with our NRLs. These highly expressed immune checkpoints indicate that patients with high risk would have more obvious drug responses and benefit from potential immunotherapy.

Besides this, we calculated the drug sensitivity under different risks. The results revealed the IC50 of AP.24534, Bosutinib, NVP.BEZ235, Rapamycin, Sunitinib, and Temsirolimus were lower in the high-risk group, while the IC50 of Erlotinib and Sorafenib was lower in the low-risk group (Figure 11A–H).

## 4. Discussion

KIRC is the most common malignant tumor in the kidney, and advanced KIRC would bring a poor prognosis. An increasing number of studies have found that necroptosis and lncRNA play an essential role in the occurrence and development of many cancers [13,24,25,26]. Researchers have constructed some NRLs signatures to predict the prognosis in different cancers, such as stomach glandular cancer [27], breast cancer [28,29], lung adenocarcinoma [30], head and neck squamous cell carcinoma [31], and colon cancer [32]. However, there still needs to be a report on NRLs to predict the prognosis of KIRC.

We received 82 DEGs from TCGA, and these genes are mainly in necroptosis and the NOD-like receptor signaling pathway. As is known, necroptosis is associated with cell death and the release of inflammation [33,34]. Nevertheless, it is noteworthy that NOD-like receptors (NLRs) have been reported as master regulators of inflammation and are involved in inflammation-related tumorigenesis, angiogenesis, and chemoresistance [35,36]. Peng Liu et al. found that aberrant activation of NLRs could occur in various cancers and blocking NLR inflammasome activation may prevent cancer progression [36]. Since these necroptosis-related DEGs are associated with tumors, co-expressed lncRNA may also play an important role. However, the role of lncRNA in KIRC may be complex. Several researchers pointed out that lncRNA SNHG16 could promote renal cell carcinoma migration and invasion by suppressing the CDKN1A [37]. However, others detected that lncRNA MAGI2-AS3 could inhibit progression and angiogenesis by interaction with transcription factor HEY1 to regulate ACY1 in KIRC [38]. The role of each lncRNA in KIRC is different and needs further verification.

To better predict the prognosis, we explored the NRLs to establish a signature for KIRC. This study applied univariate Cox regression and LASSO analysis to select which NRLs were prognosis-related and identified 6 NRLs (AC124854.1, AL117336.1, DLGAP1-AS2, EPB41L4A-DT, HOXA-AS2, LINC02100) to build a signature through multivariate Cox regression analysis. We obtained the 6 NRLs through necroptosis-related DEGs and also reverse-searched for co-expressed mRNAs of the 6 NRLs. As expected, these mRNAs are closely related to necroptosis. Among co-expressed mRNAs, TLR3 could induce necrotic death via TRIF, RIPK3, and MLKL [39], and the down-regulated of EZH2 shows a correlation with the upregulation of RIPK1 and RIPK3 [40]. IFN-β-induced necroptosis in macrophages could lead to persistent expression of STAT2 and IRF9 [41]. The necroptosis-related DEGs, 6 NRLs, and co-expressed mRNAs have formed a network closely related to necroptosis. In addition, this model reclassified the patients according to risk scores, and there were apparent differences in the clinical features between the two subtypes. All the above results can confirm the validity of this model, and the risk score can act as an independent prognostic indicator in KIRC.

KEGG and GO analysis only focus on the DEGs, while GSEA could help us observe the enrichment of gene sets from a macro perspective. In our study, the tumor and immune-associated pathways were active in the high-risk group. Chromosomal instability could cause the chronic leakage of DNA within tumor cells, then trigger a chronic inflammatory response that allows them to migrate to distant organs through a cytosolic DNA response [42]. Disorders in cytokines, their receptors or downstream signaling molecules, and increased cytokine expression are extensively involved in cancer [43]. Moreover, interleukins and chemokines could induce cancer cells to remodel the local microenvironment to work against the human body, support the growth and invasion of primary tumors and enhance metastatic colonization in many tumors [44]. These mechanisms explain why the high-risk patient has a worse outcome.

Tumor immune cells infiltrated in cancer could control and promote tumor development, growth, and invasion via remodeling the tumor microenvironment [45]. Braun DA reported that the most advanced KIRC had highly CD8+ T cell infiltrated, and only 27% had no infiltration phenotype [46]. Similarly, Su S found that the higher stage and grade of the tumor, the higher percentage of CD8+ T cells in the tumor microenvironment [47]. Moreover, the infiltration number of Tfr cells increased with the progression of lymphoma in the various pathological stages [48]. Furthermore, the degree of tumor-associated macrophage infiltration negatively correlated with the prognosis in advanced thyroid cancer [49]. In the high-risk group, the infiltration of CD8+ T cells, macrophages, T helper cells, Tfh, Th1 cells, and TIL are significantly higher, and checkpoint, inflammation-promoting, T cell co-inhibition, and co-stimulation are also. The activated immunological reaction brings a lousy prognosis to patients. However, those immune functions are essential in regulating immune responses during cancer development and treatment [50], indicating the potential therapeutic target of KIRC. In the latest clinical practice guideline for KIRC, preferred regimens include the monoclonal antibody that selectively binds to PD-1 or CTLA-4, multitargeted TKI/angiogenesis inhibitor of VEGFRs, and inhibitor of the mTOR protein [51]. Sorafenib, Sunitinib, and Temsirolimus are in this guideline; others are in the same category as first-line drugs, and future clinical trials on drug efficacy are needed. Given these findings, these small molecule compounds may be potential therapeutics for KIRC, which helps explore individualized therapeutic strategies appropriate for the patient.

We also noticed another article about the necroptosis-associated lncRNA model in KIRC [52]. The two papers have similarities in some analysis methods, but there are still differences in details. First, we compared the differences between KIRC patients and ordinary people, but they searched for the differences between the two KIRC subtypes. Secondly, the two models included different necroptosis-related lncRNAs. Our signature contained fewer lncRNAs but had a larger AUC. In addition, we also analyzed co-expression relationships between the necroptosis-related lncRNA and mRNA. Finally, the medicines obtained from anti-tumor drug sensitivity analysis in our article are supported by clinical treatment guidelines and are more reliable.

Nevertheless, it is undeniable that our study is imperfect. First, we did not use external data from other databases to verify the applicability of the NRLs signature. There are almost no data on NRLs research in KIRC and the only sequencing data in the GEO database need more patient survival information. Second, the NRLs signature is only supported by bioinformatic analysis, and experiments still need to clarify the mechanism of the NRLs in KIRC.

In general, the NRLs signature has excellent predictive performance to evaluate the prognosis in KIRC and provides the possible biological role and response for clinical treatment, but further experiments are still needed to reveal the mechanism in the future.

## Figures and Tables

**Figure 1 cells-12-00066-f001:**
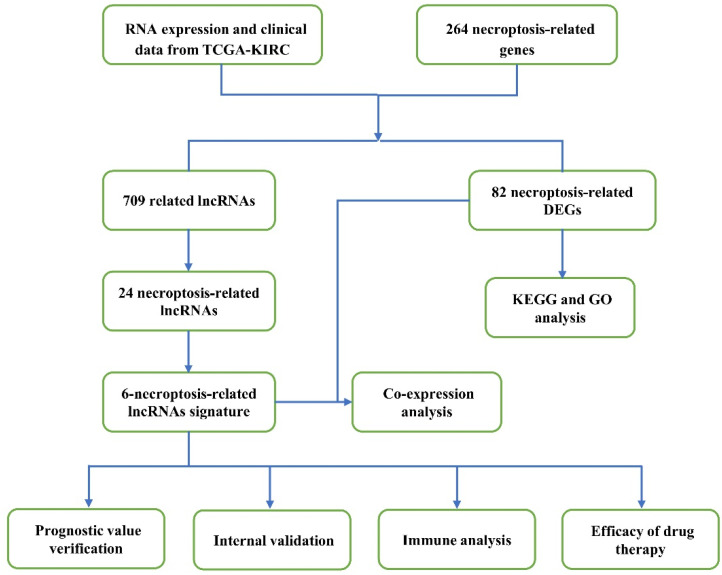
The detailed flow chart of this research.

**Figure 2 cells-12-00066-f002:**
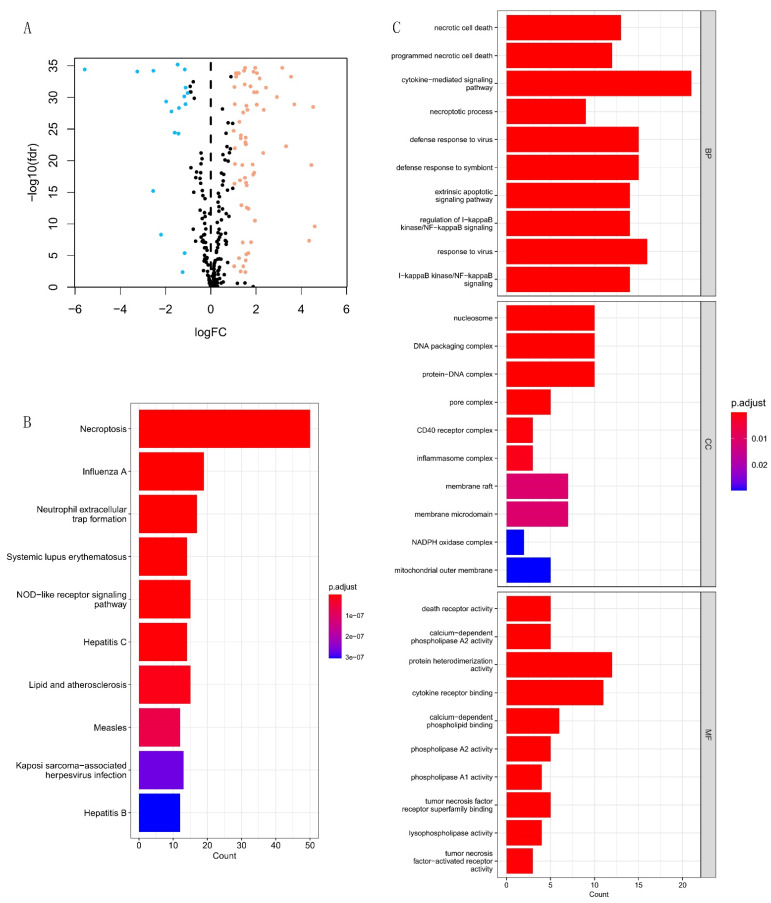
Identification of necroptosis-related DEGs between the normal and KIRC. (**A**) Volcano plot of necroptosis-related DEGs. The up-regulated and down-regulated DEGs are respectively highlighted in orange and blue. (**B**,**C**) The KEGG and GO analysis results of necroptosis-related DEGs.

**Figure 3 cells-12-00066-f003:**
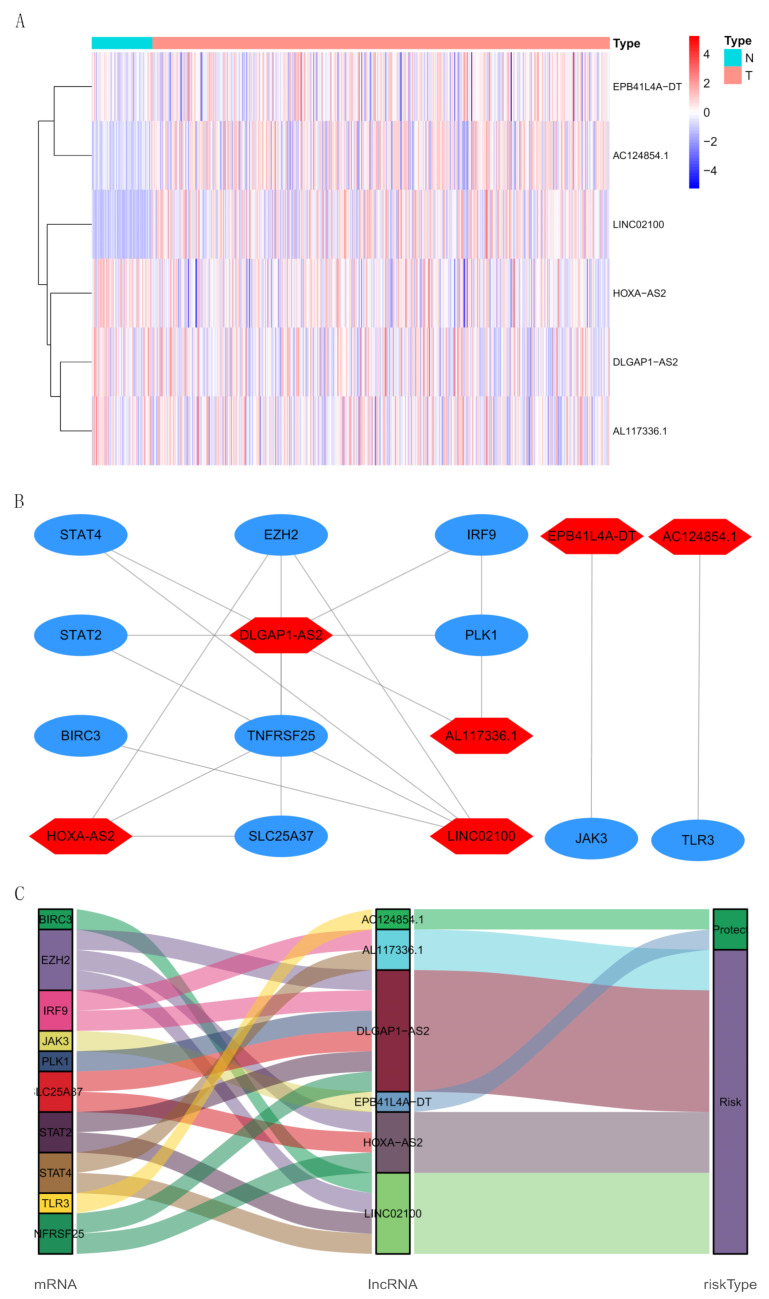
The expression level and co-expression network of 6 NRLs. (**A**) The expression level of 6 NRLs in each sample of different types. N: Normal; T: Tumor. (**B**) The co-expression relationship between 6 NRLs and mRNAs. The lncRNA and mRNA are respectively highlighted in red and blue, and the line means there is a co-expression relationship. (**C**) The interconnection of 6 NRLs, 10 mRNAs, and risk types.

**Figure 4 cells-12-00066-f004:**
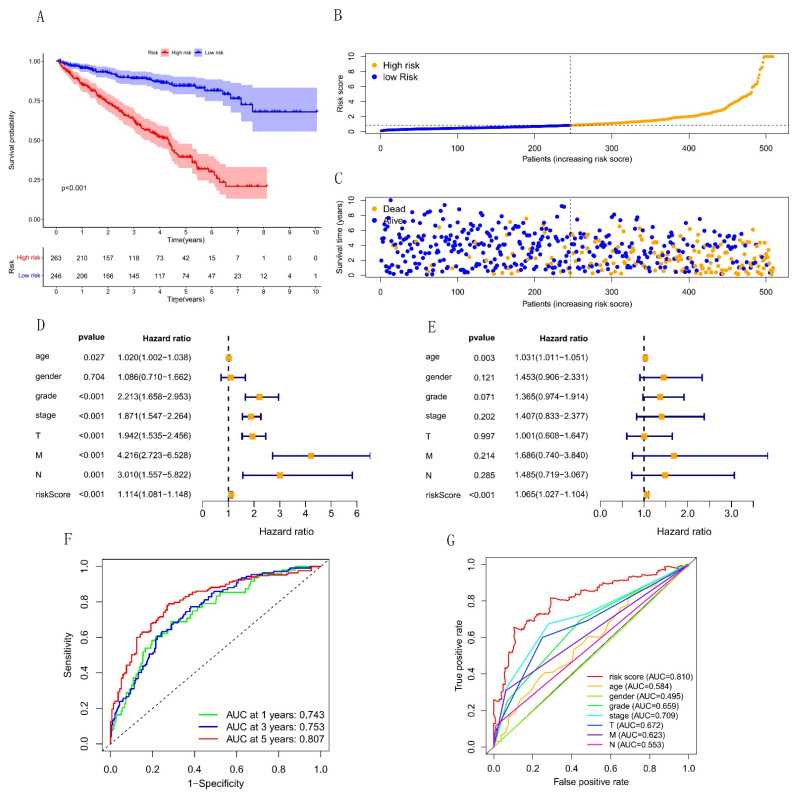
The predictive value of the risk score in prognosis for KIRC. (**A**) The survival probability in different risk groups. The high-risk and low-risk groups are respectively given in red and blue. (**B**) The distribution of the risk score among KIRC patients. Orange and blue respectively mean the high and low risk. (**C**) Changes in the number of deaths and survivors with the increase of risk score. Blue and orange respectively represent the number of survivors and deaths. (**D**–**E**) Forest plot for univariate and multivariate Cox regression analysis. (**F**) The ROC curve of the risk score at 1-year, 3-year, and 5-year survival. (**G**) The ROC curve of the risk score and clinicopathological features.

**Figure 5 cells-12-00066-f005:**
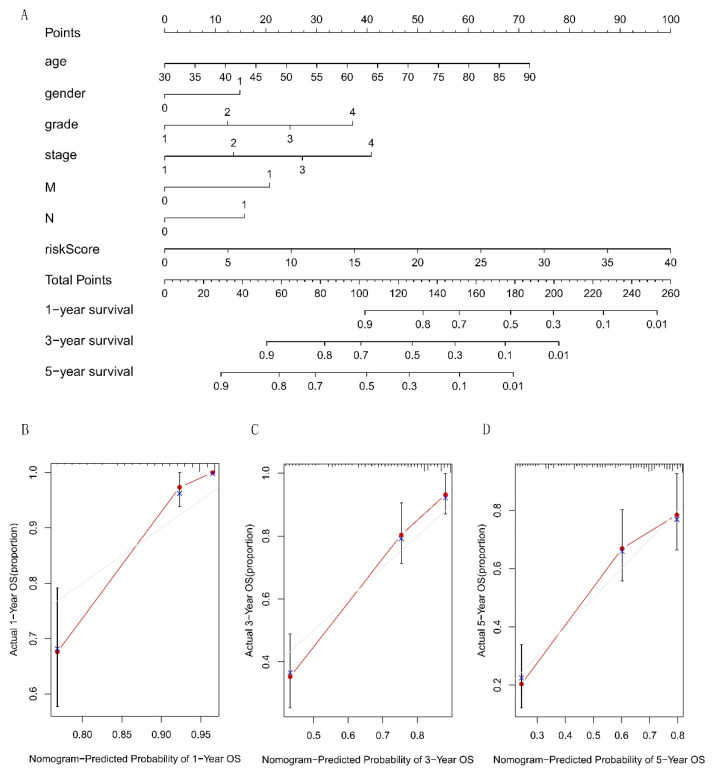
Construction and verification of the nomogram. (**A**) A nomogram contains the risk score and different pathological factors and can calculate the survival rate at 1, 3, and 5 years of KIRC patients. (**B**–**D**) The consistency test between actual OS and predicted survival at 1, 3, and 5 years indicates that the nomogram has a good prediction ability.

**Figure 6 cells-12-00066-f006:**
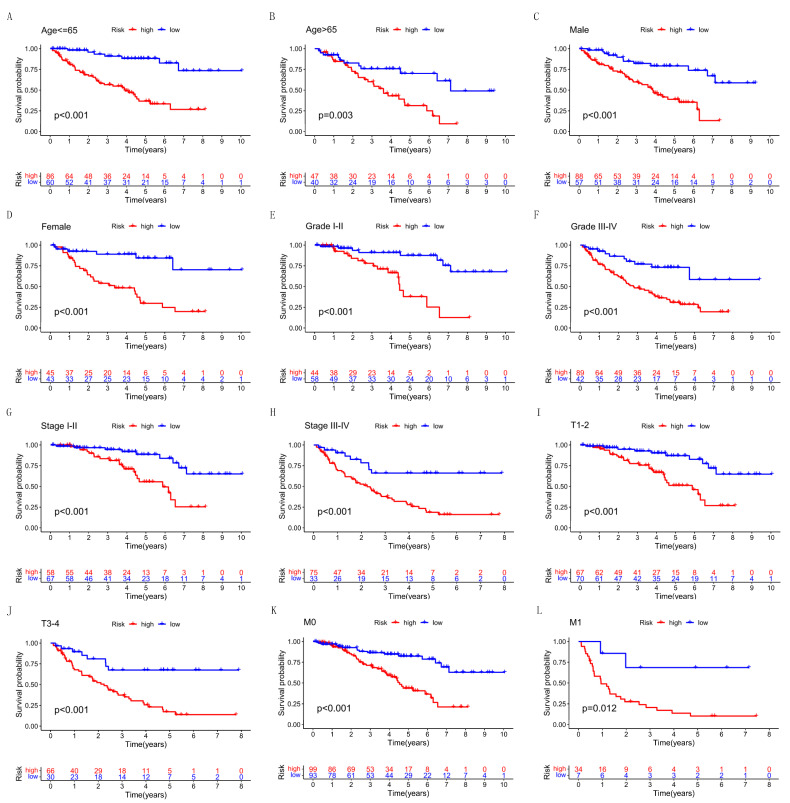
Comparison of survival rate between the high- and low-risk patients under different pathological variables. (**A**,**B**) Age. (**C**,**D**) Gender. (**E**,**F**) Grade. (**G**,**H**) Stage. (**I**,**J**) T stage. (**K**,**L**) M stage.

**Figure 7 cells-12-00066-f007:**
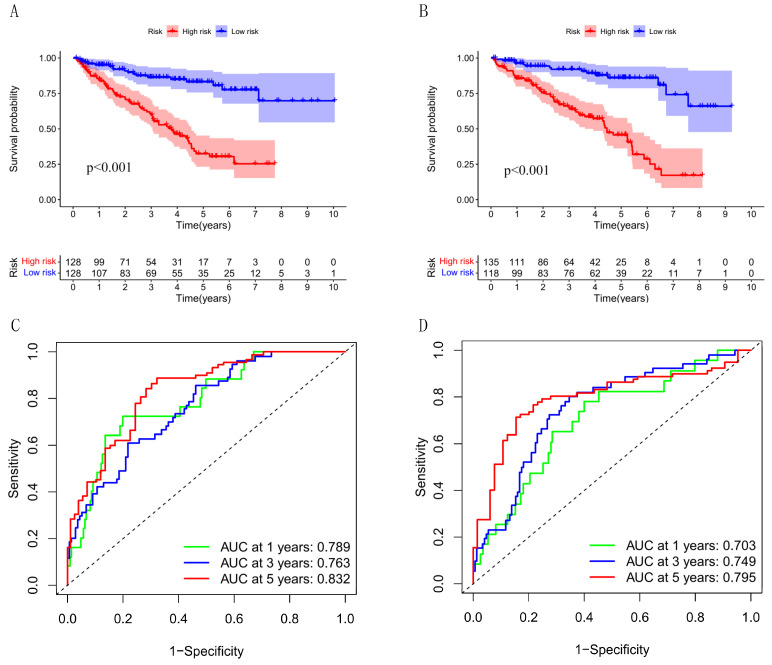
Internal validation of the risk signature. (**A**,**B**) Comparison of survival rate between high and low risk groups in internal cohorts. (**C**,**D**) The AUC at 1, 3, and 5-years survival in internal cohorts.

**Figure 8 cells-12-00066-f008:**
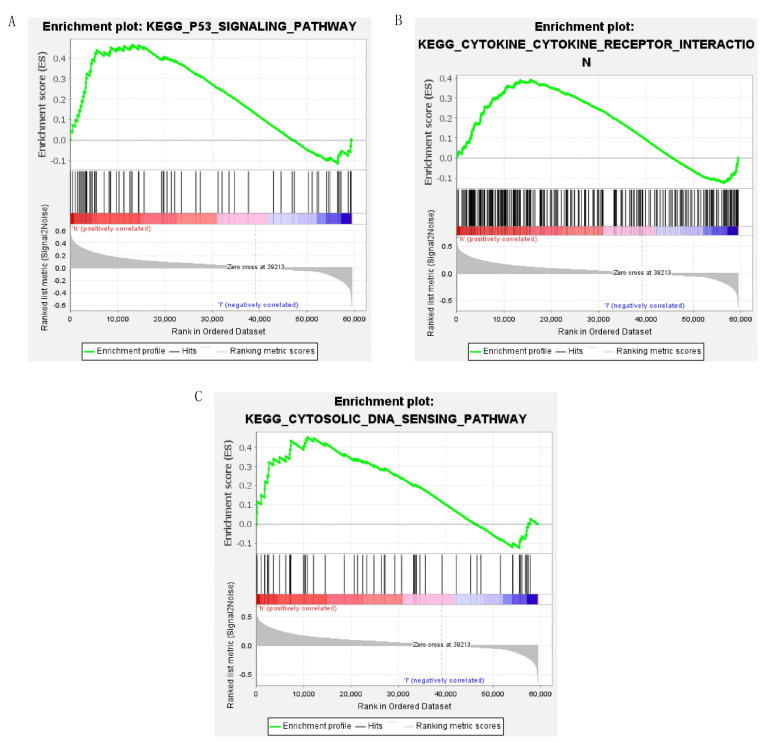
The results of GSEA in the high-risk group. (**A**) The enrichment plot of p53 signaling pathway; (**B**) The enrichment plot of cytokine-cytokine receptor interaction; (**C**) The enrichment plot of cytosolic DNA-sensing pathway.

**Figure 9 cells-12-00066-f009:**
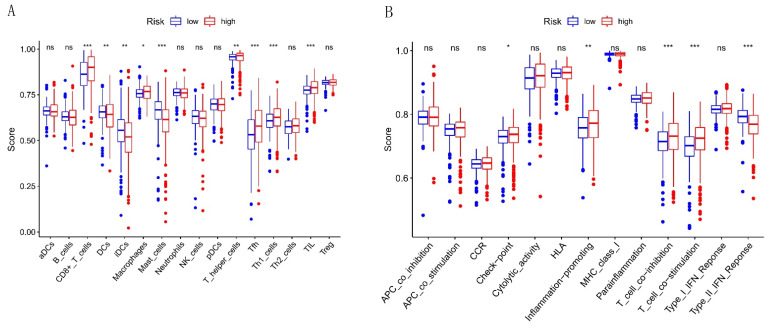
The immune difference between both risk groups. (**A**) The infiltration score of 16 immune cells. (**B**) The infiltration level of 13 immune functions. ns: non-significant; * *p* < 0.05; ** *p* < 0.01; *** *p* < 0.001.

**Figure 10 cells-12-00066-f010:**
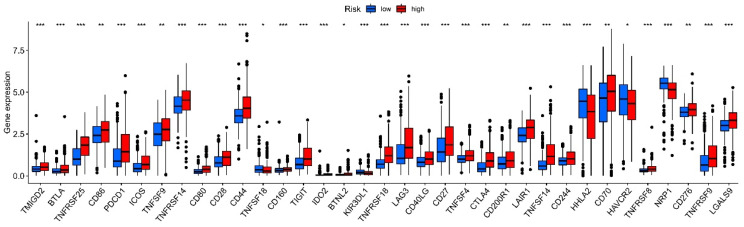
The expression of multiple immune checkpoints. * *p* < 0.05; ** *p* < 0.01; *** *p* < 0.001.

**Figure 11 cells-12-00066-f011:**
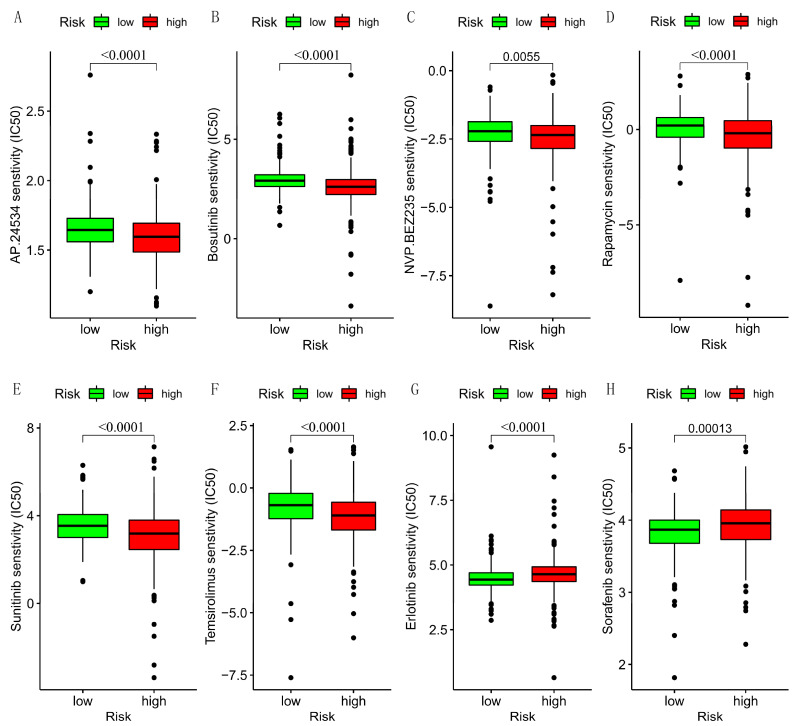
The drugs sensitivity under different risk groups. (**A**–**F**) The IC50 of AP.24534, Bosutinib, NVP.BEZ235, Rapamycin, Sunitinib, and Temsirolimus. (**G**,**H**) The IC50 of Erlotinib and Sorafenib.

**Table 1 cells-12-00066-t001:** The clinical characteristics of KIRC patients in different cohorts.

Features	Entire Cohort (*n* = 509)	Validation Cohort	
		First Cohort (*n* = 256)	Second Cohort (*n* = 253)
Age (%)			
≤65	337 (66.2)	167 (65.2)	170 (67.2)
>65	172 (33.8)	89 (34.8)	83 (32.8)
Gender (%)			
Female	175 (34.4)	89 (34.8)	86 (34.0)
Male	334 (65.6)	167 (65.2)	167 (66.0)
Grade (%)			
G1	12 (2.4)	8 (3.1)	4 (1.6)
G2	216 (42.4)	110 (43.0)	106 (41.9)
G3	199 (39.1)	95 (37.1)	104 (41.1)
G4	74 (14.5)	39 (15.2)	35 (13.8)
GX + unknown	8 (1.6)	4 (1.6)	4 (1.6)
Stage (%)			
Stage I	254 (49.9)	127 (49.6)	127 (50.2)
Stage II	53 (10.4)	29 (11.3)	24 (9.5)
Stage III	116 (22.8)	51 (20.0)	65 (25.7)
Stage IV	83 (16.3)	48 (18.7)	35 (13.8)
unknown	3 (0.6)	1 (0.4)	2 (0.8)
T (%)			
T1	260 (51.1)	132 (51.6)	128 (50.6)
T2	65 (12.8)	35 (13.7)	30 (11.9)
T3	173 (33.9)	84 (32.8)	89 (35.2)
T4	11 (2.2)	5 (1.9)	6 (2.3)
M (%)			
M0	402 (79.0)	194 (75.8)	208 (82.2)
M1	79 (15.5)	47 (18.4)	32 (12.6)
MX + unknown	28 (5.5)	15 (5.8)	13 (5.2)
N (%)			
N0	226 (44.4)	119 (46.5)	107 (42.3)
N1	16 (3.1)	9 (3.5)	7 (2.8)
NX	267 (52.5)	128 (50.0)	139 (54.9)

T, tumor; M, metastasis; N, lymph node.

## Data Availability

The datasets presented in this study can be found in TCGA. The names of the repository can be found in the article. Further inquiries can be directed to the author.

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
