# Peer review of "Construction of a Necroptosis-Related lncRNA Signature for Predicting Prognosis and Immune Response in Kidney Renal Clear Cell Carcinoma"

_cells, 2022, doi:10.3390/cells12010066_

Round 1

Reviewer 1 Report (Previous Reviewer 2)

The authors much improved the manuscript, but still must have it checked for English language. I have no further objection to the publication of this manuscript

Author Response

I have completely revised the English language of this article. Thank you for your valuable comments.

Reviewer 2 Report (New Reviewer)

Dear Authors,

The paper submitted describes a set of lncRNAs that could be used to predict the prognosis of KIRC patients independently of their histopathological features. The strategy to abord the problematic it´s appropriate and leads to acceptable lncRNA signature related to the Necroptosis program. However, the presentation of your results must be improved. The size of the font in several graphic representations it too small and makes it difficult to read and understand the figure. The figure legends should be more informative i.e Fig 3 and 5 are very poorly explained. Apparently, the entire legend of fig 4 is missing, making it difficult to understand what you are presenting in the results and late discussion.

Finally, you should revise this paper and include their findings in your formulation and discussion, because both works are very similar.

Gu J, He Z, Huang Y, et al. Clinicopathological and Prognostic Value of Necroptosis-Associated lncRNA Model in Patients with Kidney Renal Clear Cell Carcinoma. Dis Markers. 2022;2022:5204831. Published 2022 May 23. doi:10.1155/2022/5204831

In the next version of your submission please upload a definitive file and not a revised version with “track changes” mode, which is very confusing to read. Also, you should highlight the differences between the two works.

Author Response

1) I have completely revised the English language of this article.

2) We recreated high-pixel figures to ensure the readability.

3) We revised the figure legends.

4) We explained the differences between the two articles in the discussion (Line 339-347). The two papers have similarities in some analysis methods, but there are still differences in details. First, we compared the differences between KIRC patients and ordinary people, but they searched for the differences between the two KIRC subtypes. Secondly, the two models included different necroptosis-related lncRNAs. Our signature contained fewer lncRNAs but had a larger AUC. In addition, we also analyzed co-expression relationships between the necroptosis-related lncRNA and mRNA. Finally, the medicines obtained from anti-tumor drug sensitivity analysis in our article are supported by clinical treatment guidelines and are more reliable. I sincerely appreciate your valuable suggestions and comments.

Round 2

Reviewer 2 Report (New Reviewer)

Dear Authors, in my opinion, this paper has been done well and is acceptable but it lacks of empiric results, principally related to efficiency of drug therapy.

This manuscript is a resubmission of an earlier submission. The following is a list of the peer review reports and author responses from that submission.

Round 1

Reviewer 1 Report

The authors provide an elaborate study assessing the role of necroptosis, associated pathways and related gene-signatures, in clear cell renal cell carcinoma. The study is extensive, with an abundance of data - some of which is hypothesis generating. Necroptosis is certainly an interesting cellular process that requires further investigation for cancer therapeutics. I commend the authors on this novel study, however, the manuscript as well as some methodologies limit its applicability and overall value. The manuscript itself is has significant grammar issues, with illegible writing at times. This, in and of itself, prohibits publication. Moreover, some of the modelling and study design as little clinical relevance. To this end, this study is not suited for publication. Some specifics below:

1)      Introduction is difficult to understand – both due to grammar, poor language proficiency, and overall organization. Necroptosis and the intended study rationale needs to be much clearer. Lots of molecular and genetic drivers discussed with no coherent flow. Is there a role for epigenetics here with the LncRNA? Its seems to be suggested yes, but not clear.

2)      Line 44: Survival data for mRCC is out of date. Novel studies have transformed the field significantly (see landmark combination therapy trials from 201 – present).

3)      Materials and methods: necroptosis is such a complex process! As the authors point out, there are inducers and inhibitors of the process – all of which are “necroptosis-related genes”. Of the 264 necroptosis related genes identified, was specific function accounted for? Or was everything included that was designated as part of the necroptosis process.

4)      Assessment and prognostic signature (line 101): I think its fine to define ROC and KM survival analyses – but definitions and imprecise language (ie: “Usually used”) do not belong here.

5)      The expression of Immune checkpoints is NOT predictive to ICI response in most large studies (aside from a few like CM-214). I caution this implication that expression levels are clinically relevant.

6)      Figure 2C, y-axis font should be larger (eg. BP, CC, MP)

7)      Line 161: protective factors for what? Does this mean associated with longer OS?

8)      Figure 4: this data set requires a table of patient demographics and characteristics. Is this available through your dataset? We currently have a predictive model in RCC that is validated (IMDC). The table should include IMDC risk, hx of nephrectomy, sarcomatoid differentiation on histology. I am assuming this is clear cell patients only based on the title of the study, but this should be made clearer in the manuscript.

9)      Can you elaborate on the dataset? Were these patients all treatment-naïve? Or had they received prior treatment. What is survival based on? If data is available on what treatments were given and how many lines, that also belongs in the aforementioned table.

10)   Figure 4b, looks like a minority of patients are driving poor survival in the high risk group (patients clustered at the far right of the graph). Please explain why the low vs high risk demarcation was where it was. Could you consider defining high risk differently?

11)   Figure 4E: so disease stage and presence of metastases were NOT significant here?! How can this be? These are by virtue risk factors for death. This implies there is an issue with the validity of the dataset.

12)   The nomogram is very interesting. However, again, its value nests in knowing if these patients were placed on therapy. This is a huge factor and can be a confounder.

13)   Figure 6 is remarkable, and very convincing. Recommend further elaboration on these results with actual text beyond what is written in lines 198-201

14)   Figure 7 p values can be simplified (<0.05 ect)

15)   Lines 225-233 belong in discussion, not results.

16)   GSEA and immunity analysis, what is meant by p53 signaling pathway analysis? Is this patients with mutation and aberrant function?

17)   Immune data from line 244, some of this belongs in the discussion

18)   Figure 9: Would be nice to also add data on suppressor immune cells, since they are relevant to immunogenicity of RCC and the tumor microenvironment.

19)   There is absolutely no role for conventional chemotherapy in the current treatment paradigm of clear cell mRCC outside of some trials. This is not standard of care. Where is this data with bleomycin, doxorubicin, and crizotinib coming from? What patients were these? This is not relevant.

20)   Discussion is largely superficial. There is so much data here (you should be proud!), but it needs to be tied together. Introduction and discussion are lacking and remain unclear.

21)   Limitations belong in discussion, not conclusion. Needs a stronger conclusion.

Author Response

Point 1: Introduction is difficult to understand – both due to grammar, poor language proficiency, and overall organization. Necroptosis and the intended study rationale needs to be much clearer. Lots of molecular and genetic drivers discussed with no coherent flow. Is there a role for epigenetics here with the LncRNA? Its seems to be suggested yes, but not clear.

Response 1: I rewrote the introduction and further clarified the purpose of this study.

Point 2: Line 44: Survival data for mRCC is out of date. Novel studies have transformed the field significantly (see landmark combination therapy trials from 201 – present).

Response 2: The survival data was only to emphasize the poor prognosis of patients in the past. And in this paper we mentioned although immunotherapy and target therapy have certain therapeutic effects nowadays, the outcome of patients in the late-stage still remains disma.

Point 3: Materials and methods: necroptosis is such a complex process! As the authors point out, there are inducers and inhibitors of the process – all of which are “necroptosis-related genes”. Of the 264 necroptosis related genes identified, was specific function accounted for? Or was everything included that was designated as part of the necroptosis process.

Response 3: These genes are involved in the necroptosis pathway and related to necroptosis, with no specific function.

Point 4: Assessment and prognostic signature (line 101): I think its fine to define ROC and KM survival analyses – but definitions and imprecise language (ie: “Usually used”) do not belong here.

Response 4: I restated the definition of ROC and KM survival analyses.

Point 5: The expression of Immune checkpoints is NOT predictive to ICI response in most large studies (aside from a few like CM-214). I caution this implication that expression levels are clinically relevant.

Response 5: The expression of immunological checkpoints in different risk groups may only be related to ICI response, but this is certainly not absolute.

Point 6:  Figure 2C, y-axis font should be larger (eg. BP, CC, MP)

Response 6: I have changeed the y-axis font.

Point 7: Line 161: protective factors for what? Does this mean associated with longer OS?

Response 7: The protective factor means it is beneficial to patients and indicates a longer OS.

Point 8: Figure 4: this data set requires a table of patient demographics and characteristics. Is this available through your dataset? We currently have a predictive model in RCC that is validated (IMDC). The table should include IMDC risk, hx of nephrectomy, sarcomatoid differentiation on histology. I am assuming this is clear cell patients only based on the title of the study, but this should be made clearer in the manuscript.

Response 8: We used the information in TCGA database to supplement a table (table 1), which contained the clinical characteristics of KIRC patients in different cohorts.

Point 9: Can you elaborate on the dataset? Were these patients all treatment-naïve? Or had they received prior treatment. What is survival based on? If data is available on what treatments were given and how many lines, that also belongs in the aforementioned table.

Response 9: Our data comes from the open database (TCGA). When sequencing, the patient did not receive radiotherapy or chemotherapy, but TCGA did not provide follow-up treatment information for the patient. All patient survival is based on overall survival.

Point 10: Figure 4b, looks like a minority of patients are driving poor survival in the high risk group (patients clustered at the far right of the graph). Please explain why the low vs high risk demarcation was where it was. Could you consider defining high risk differently?

Response 10: We calculate the risk score of each patient according to the risk model, and then display it in figure 4b from low to high. The fact that patients gather at the far right of the figure only shows that their risk scores are relatively high, but not that their survival rate is low. The division of high and low risks is based on the median of risk scores of all patients. Those below the median are low risks, while those above the median are high risks. This method is commonly used in most articles at present.

Point 11: right of the graph). Please explain why the low vs high risk demarcation was where it was. Could you conside

Response 11: Disease stage and metastasis are indeed important risk factors for death. We have confirmed this through univariate COX analysis and the results are shown in Figure 4D. In Figure 4E, we used the Multivariate COX analysis, and it only showed disease stage and metastasis cannot be independent predictors in this study, but the risk score could.

Point 12: The nomogram is very interesting. However, again, its value nests in knowing if these patients were placed on therapy. This is a huge factor and can be a confounder.

Response 12: We use the nomogram to predict the survival rate of patients, which cannot reflect whether the patients are on therapy.

Point 13: Figure 6 is remarkable, and very convincing. Recommend further elaboration on these results with actual text beyond what is written in lines 198-201

Response 13: We explained these results in detail in the discussion.

Point 14: Figure 7 p values can be simplified (<0.05 ect)

Response 14: I don't think it is necessary to modify.

Point 15: Lines 225-233 belong in discussion, not results.

Response 15: We have accepted your suggestion and made changes.

Point 16: GSEA and immunity analysis, what is meant by p53 signaling pathway analysis? Is this patients with mutation and aberrant function?

Response 16: Traditional enrichment analysis can only locate the functions and show which functions are related to these differential genes, but cannot answer the overall state of a certain pathway. GSEA analysis showed that p53 signaling pathway was activated in high-risk patients. P53 can trigger apoptosis, indicating the selected necroptosis-related lncRNA is significant in this paper. The purpose of our study is not to analyze the mutation of p53 gene between high and low patients.

Point 17: Immune data from line 244, some of this belongs in the discussion

Response 17: We have accepted your suggestion and made changes.

Point 18: Figure 9: Would be nice to also add data on suppressor immune cells, since they are relevant to immunogenicity of RCC and the tumor microenvironment.

Response 18: Figure 9 contains the analysis of immunosuppressive cells (such as Treg cells and immature DCs), which you may not have noticed.

Point 19: There is absolutely no role for conventional chemotherapy in the current treatment paradigm of clear cell mRCC outside of some trials. This is not standard of care. Where is this data with bleomycin, doxorubicin, and crizotinib coming from? What patients were these? This is not relevant.

Response 19: We re analyzed these results and selected the predicted drugs in combination with the latest NCCN clinical practice guideline for KIRC.

Point 20:  Discussion is largely superficial. There is so much data here (you should be proud!), but it needs to be tied together. Introduction and discussion are lacking and remain unclear.

Response 20: We rewrote the discussion and conclusion of the article.

Point 21:  Limitations belong in discussion, not conclusion. Needs a stronger conclusion.

Response 21: We rewrote the discussion and conclusion of the article.

Reviewer 2 Report

This is an interesting study where the authors identified a long non-coding RNAs (lncDNAs) signature with prognostic and immune response value for KIRC. The results of this study could potentially be important to this field. However, I believe that the authors did not validate their findings in an appropriate manner. Indeed, the cohort should have been randomly divided into a development cohort (training and testing cohort) and internal validation cohort. Additionally, the authors should have further validated their findings using an external cohort to determine whether their prognostic model is reproducible and can be generalized to different patients. The absence of external validation was mentioned by the authors in the concluding remarks and I appreciate their straightforwardness, but it remains an important point that cannot be overlooked in my opinion.

Minor comments:

-Fig 1 indicates 83 necroptosis-related DEGs from 264 genes, but in result section line 128 it says 82 DEGs from 258 genes. I understand that the authors collected 264 necroptosis -related genes and only 258 could be found in the transcriptome data, but the discrepancy between text and figures can be confusing here.

- Would it not be possible to do a multivariate analysis of the different pathological variables of the dataset, including the lncDNA signature, to see how it plays out?

Author Response

Point 1: Fig 1 indicates 83 necroptosis-related DEGs from 264 genes, but in result section line 128 it says 82 DEGs from 258 genes. I understand that the authors collected 264 necroptosis -related genes and only 258 could be found in the transcriptome data, but the discrepancy between text and figures can be confusing here.

Response 1: In Figure 1, there should be 82 necroptosis-related DEGs, which were wrongly labeled due to my negligence. I have corrected the error mark.

Point 2: Would it not be possible to do a multivariate analysis of the different pathological variables of the dataset, including the lncDNA signature, to see how it plays out?

Response 2: In this paper, we constructed a nomogram, including different pathological characteristics (such as age, sex and different pathological stages) and risk scores of the lncRNA signature, to comprehensively predict the 1-year, 3-year and 5-year survival rates of patients.

In addition, we use TCGA database to build and validate this model, because the sequencing data and survival information are relatively complete. However, the research on KIRC in GEO and other databases lack survival data of patients, so it is impossible to find an appropriate external cohort for verification. I have referred to many other articles and they also lack external data validation, which may be related to the difficulty in obtaining external data.
